# Evidence for motor imagery in the management of vestibular disorders does not support recent guidelines: A systematic search and review

Dimitri Fabre-Adinolfi[1], Florian Naye[2], Thomas Rulleau[3]*

**1** Institut Lorrain de Formation en Masso-Kinésithérapie de Lorraine, Nancy, France, **2** Université de Sherbrooke, Faculty of Medicine and Health Sciences, School of Rehabilitation, Research Centre of the CHUS, CIUSSS de l'Estrie-CHUS, Sherbrooke, Quebec, Canada, **3** Nantes Université, CHU Nantes, Direction de la Recherche et de l'Innovation, Movement - Interactions - Performance, MIP, UR 4334, Nantes, France

* thomas.rulleau@chu-nantes.fr

## Abstract

### Context

Vestibular disorders significantly impact people's quality of life, affecting functions such as balance and walking. Current treatments involve pharmacological and non-pharmacological approaches, with vestibular rehabilitation proving effective in helping to compensate for deficits. Although traditional techniques address various aspects of the vestibular system, many fundamental studies suggest motor imagery could reduce vestibular impairment. Despite the paucity of research on motor imagery for vestibular disorder rehabilitation, recent expert recommendations suggest it could enhance functional recovery. This review questions the evidence in the literature on the clinical value benefits of motor imagery as part of vestibular rehabilitation. The aims of this study were 1. to identify motor imagery interventions used in the literature, 2. to critically appraise them, and 3. to report their impact on clinical outcomes.

### Methods

We conducted a systematic search using the PubMed, Cochrane Library, Web Of Science, CINAHL and Scopus databases. The review was registered on PROSPERO and is reported according to the PRISMA guidelines. PICO criteria were adults undergoing a motor imagery intervention as part of vestibular rehabilitation, studies with or without comparators, and the use of clinical outcome measures. A critical grid was completed, and a risk of bias assessment was performed.

### Results

Despite identifying 2404 (after duplicate removal) concerning motor imagery and vestibular rehabilitation, only two articles met our inclusion criteria. Our findings suggest

**Data availability statement:** We have collected no data for this paper.

**Funding:** The author(s) received no specific funding for this work.

a clinical benefit of integrating motor imagery into vestibular rehabilitation. However, the limited evidence and methodological shortcomings warrant caution, including small sample size, absence of imagery quality assessment, and poor generalizability.

## Conclusion

Future research should address the identified limitations, including the need to study a broader range of vestibular pathologies, use comprehensive assessments, and evaluate long-term effects. This would contribute to a more thorough understanding of motor imagery as part of vestibular rehabilitation.

## Trial registration

PROSPERO CRD42023444673

## Introduction

People living with vestibular disorders report a considerable impact on their quality of life [1]. Vestibular disorders affect multiple functions (e.g., balance and gait), leading to incapacity and participation restriction [2]. Current treatments for vestibular disorders include pharmacological and non-pharmacological approaches [3,4]. Vestibular rehabilitation is an effective and specialized approach that aims to promote compensation for vestibular deficits, improve balance, reduce dizziness and vertigo symptoms, and enhance overall functional independence [5]. Vestibular rehabilitation techniques encompass a range of exercises and interventions targeting different aspects of the vestibular system [6,7]. Gaze stabilization exercises improve coordination between the vestibular and visual systems. Balance training exercises focus on postural control and stability. Habituation exercises gradually expose individuals to movements or stimuli that provoke their symptoms [8]. All these techniques enhance overall balance and functional performance. Recently, motor imagery has emerged as a promising technique targeting the cognitive processes involved in movement generation [9–11].

Motor imagery is based on the simulation theory [12,13], which posits that every action consists of two phases: an internal, invisible phase of anticipation, and a visible phase of execution [12]. The anticipation phase includes the purpose of the action, the meaning of the action and the consequences of the action on the organism and the outside world [13]. The simulation theory postulates that the anticipation phase involves all the neural mechanisms of an action that is internally generated but not physically executed [13]. Accordingly, motor imagery refers to the mental simulation of a movement without actual execution [9,11], a process that relies on intact sensory and cognitive systems. Among these, vestibular sensory processing both influences and is influenced by higher cognitive functions. Yardley et al. showed that vestibular pathology can cause interference between postural control and the execution of mental tasks in people with vestibular disorders [14]. Schönherr and May showed that caloric vestibular stimulation affects the perceptual component of body experience in

healthy participants, highlighting its role in modulating the different perceptual qualities that contribute to our body experience [15]. In an experimental study, Zhang et al. [16] demonstrated specific vestibular activity after a program based on motor imagery and vestibular stimulation motor imagery. Pérush et al. (2011) also shows that vestibular damage disorganizes brain structures commonly involved in mental imagery, and more generally in mental representation [17]. Mast et al. 2005 show that vestibular stimulation affects mental imagery and mental rotation [18]. All of these studies highlight the impact of vestibular dysfunction—whether experimentally induced or pathological—on motor imagery abilities. They also raise the question of a possible bidirectional relationship between the vestibular system and motor cognitive processes.

Experts have made recommendations for using motor imagery in vestibular rehabilitation, but no literature has been cited supporting them. A recent narrative review [10] recommended integrating imagined body movements (i.e., motor imagery) into a vestibular rehabilitation protocol to improve functional recovery, without providing supporting evidence. A review of specific studies on this subject is therefore required. In addition, information is needed on the interventions to be applied, the specific effects and other characteristics useful to clinicians implementing these interventions and researchers aiming to better understand motor imagery in vestibular rehabilitation.

The aim of this study was to (1) identify motor imagery interventions used in the context of vestibular rehabilitation, (2) critically appraise them based on best practice elements from the fields of motor imagery and vestibular rehabilitation, and (3) synthesize their reported effects on clinical outcomes.

## Materials and methods

To achieve our objective, we conducted a "systematic search and review" [19]. This approach combines the rigorous search strategy of a systematic review with the interpretive flexibility of a critical review, making it particularly well suited to broad and exploratory questions [19]. This method allows for the inclusion of diverse study types, enabling a comprehensive overview of the existing literature [19]. Unlike traditional systematic reviews, which aim to evaluate intervention effectiveness and therefore apply tools such as GRADE to rate the certainty of effect estimates, a systematic search and review focuses on describing interventions, examining their clinical applicability, and identifying conceptual or methodological gaps [19,20]. In this context, GRADE or equivalent frameworks, which are designed to assess confidence in pooled or comparative effect estimates, are not appropriate [21]. Moreover, this approach supports the integration of expert-informed appraisal criteria [20], which we used to evaluate the clinical relevance and feasibility of identified interventions based on best practice elements in motor imagery and vestibular rehabilitation. No methodological standards were available for this approach. To ensure a robust synthesis, an adapted version of the JBI Manual for Evidence Synthesis was used to fit with the systematic search and review typology [22]. The review and results are reported according to the Preferred Reporting Items for Systematic Reviews and Meta-Analyses (PRISMA-2020). The protocol was registered in PROSPERO (CRD42023444673).

### Eligibility criteria

Table 1 presents the eligibility criteria for articles related to the research question according to the Population, Intervention, Comparator and Outcomes (PICO) framework [25].

| Population | Adults with a vestibular disorder or undergoing vestibular stimulation |
|---|---|
| Intervention | Vestibular rehabilitation with motor imagery |
| Comparator | All other interventions including no comparator |
| Outcomes | All reported clinical outcomes |

### Information sources

The systematic search was undertaken using multiple sources: Scopus, Web of Science, Cochrane Library, OVID Medline, and CINAHL.

**Table 1. Extraction grid.**

| | Study 1 (Nigmatullina et al. 2015) [23] | Study 2 (Jahn et al., 2002) [24] |
|---|---|---|
| Age | 26 (10 males/ 16 females) | 10 (8 males/ 2 females) |
| Study type | Comparative study | Self – comparative study |
| Comparison | Comparison between congruent and incongruent mental imagery tasks as a function of actual rotation | Comparison between actual standing/walking and imagined standing/walking/running 3–6 days after symptom onset |
| Pathological subjects | | |
| BPPV, Schwanomme, | No | No |
| Oscillopsias, Meniere's disease | No | No |
| Neuritis | No | Yes |
| Type of lesion | Bilateral | Unilateral |
| Healthy subject | Yes | No |
| Strict/ broad | Strict | Strict |
| Representative | No | No |
| Measurement time | Immediate | 3 to 6 days delayed |
| Intervention details (no. of repetitions/pause/time) | Trials consisted of chair rotations in two directions (left and right) under three imaging conditions | Measurement of vestibulo-ocular reflex during actual walking and standing versus 3 imagined locomotion tasks |
| Measurement instrument | VOR+SPV | VOR+SPV |
| Tool validated for this population | Yes | Yes |
| Information on sensitivity to change | Yes | Yes |
| Videonystagmographic measurement | Yes | Yes |
| Caloric test/ Rotary | Yes/ No | Yes/ No |
| Rotatory or galvanic stimulation | Yes rotatory | No |
| Other vestibular rehabilitation tools | No | No |
| Testing after vestibular disorders | Yes | Yes |
| Immediate | Yes | No |
| Deferred | No | Yes |
| Other outcomes | | |
| Questionnaires | No | No |
| Perceived balance (DHI type) | No | No |
| Quality of life (WoQol-Brief) | No | No |
| Physical activity practice | No | No |
| Balance test | No | No |
| Single limb stance | No | No |
| Stabilometry | No | No |
| Equitest | No | No |
| Statistics (p-value) | * | * |
| Data | SPV ↘* in congruent IM condition | SPV ↘* in IM running |
| Imagery protocol | | |
| Imagery quality assessment | Not evaluated | Not evaluated |
| Motor imagery training | No | No |
| Motor imagery education | No | No |
| Type of imaging | | |
| Whole body, mental rotation or Undefined | Whole body | Whole body |
| Perspective/ Modality whole body | First person/ kinesthesic | First person/ kinesthesic |

## Search strategies

An iterative process and the PRESS Checklist [26] were used to develop the search strategy. This search strategy was developed with combinations of keywords from both the domains of our study (i.e., vestibular disorders and mental imagery). This search strategy was adapted for the other databases (See Supplementary Material 1 for the full search strategy and justification of search strategy refinements following protocol registration). There were no date restrictions. All searches were performed June 1st, 2025. We performed handsearching of the reference lists of included studies to identify additional relevant citations.

## Selection process

A calibration training was performed for title and abstract screening using 10 randomly selected references. If the agreement percentage between the two reviewers was below 80% [27], the eligibility criteria were refined, and a new calibration phase was conducted.

Two reviewers (TR and DF) with expertise in the vestibular and motor imagery fields independently screened the title and abstract of each citation. The same two reviewers then screened the full texts of potential studies. Disagreement between reviewers was handled with a third reviewer (FN). The reasons for full-text exclusion (no human participants, adults, no motor imagery intervention, no vestibular rehabilitation, no vestibula desease or vestibular simulation) were documented and are reported in the PRISMA 2020 flow chart (Fig 1).

## Data collection process

Two independent reviewers (DF and RT) extracted the data from the retained studies. In case of disagreement, consensus was reached with a third reviewer (FN).

## Data items

The following study characteristics were extracted: authors, year of publication, study design, comparator(s), sample size, participant characteristics, outcomes, assessment timepoint(s), and measurement instruments.

## Critical appraisal grid

One objective of a systematic search and review is to provide practice-oriented insights rather than to appraise internal validity for causal inference [19]. Accordingly, we developed a tailored critical appraisal tool to evaluate the external validity, feasibility, and alignment of intervention components with current best practice in motor imagery and vestibular rehabilitation. This grid was designed to capture aspects of intervention content and clinical applicability that are not addressed by traditional risk-of-bias tools. Key elements of contemporary motor imagery and vestibular practice were therefore integrated to support a clinically meaningful interpretation of the included trials.

Regarding the methodological and feasibility criteria, the critical appraisal tool was composed of items relating to external validity to verify the feasibility and generalizability of the interventions tested in the included studies [28], statistical validity to avoid misinterpretation of the results of these studies [29,30], and internal validity to limit biases that could affect the cause-and-effect relationship [31]. The quality criteria for studies involving motor imagery and vestibular rehabilitation were assessed to ensure that the proposed interventions were both clinically feasible and consistent with best practice. These evaluations (i.e., motor imagery capacity evaluation, perspective, modality, familiarizations, closed eyes) include criteria that impact the quality of the image produced and the outcomes of the intervention, but also enable the intervention to be reproduced. The setting in which the rehabilitation was carried out and the type of stimulation (i.e., rotary chair, optokinetic stimulation, vestibular stimulation exercises, work on eye-motor movements, habituation exercises and balance rehabilitation work) were noted. The development of this critical appraisal tool was based on specific elements

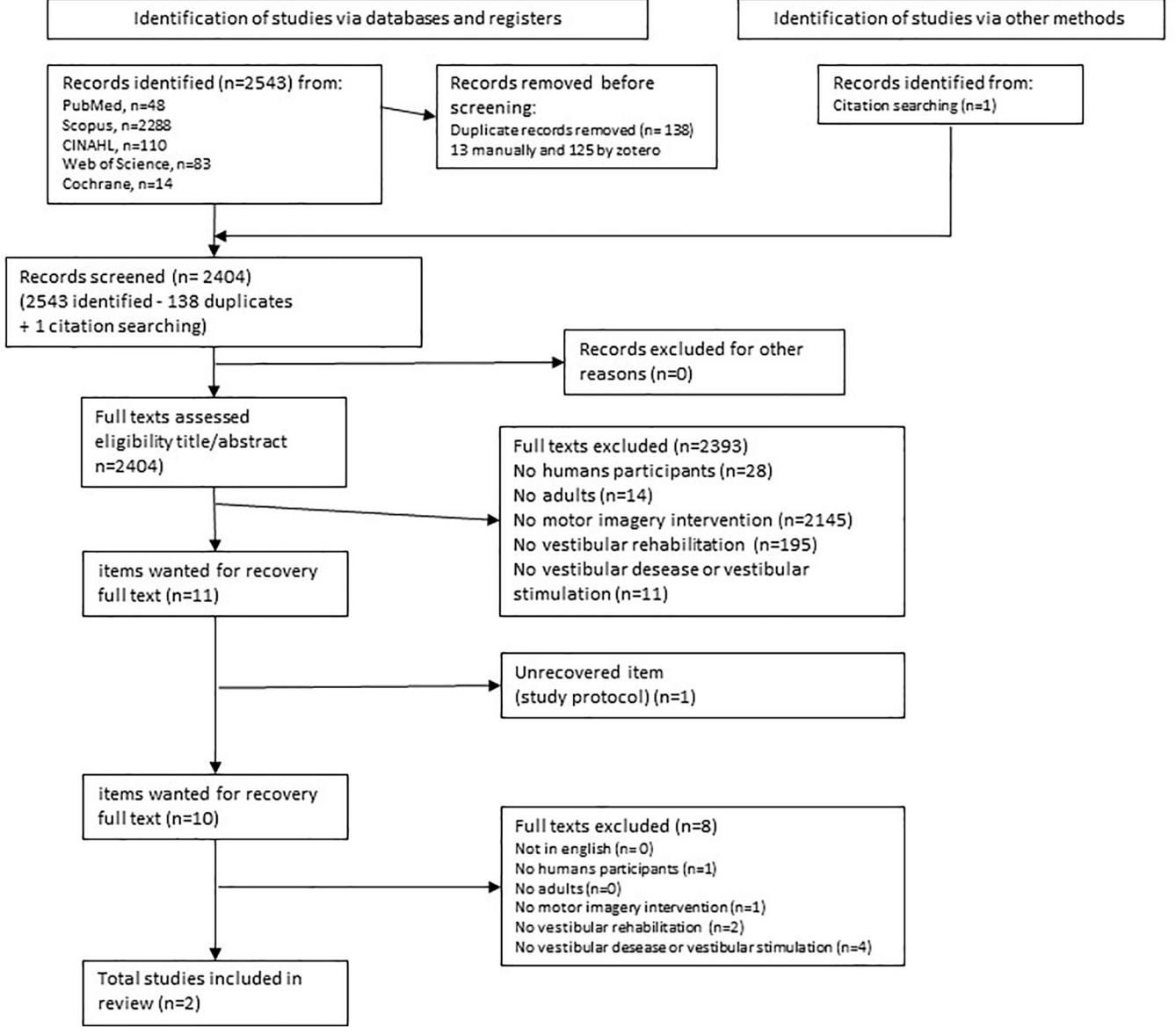

**Fig 1. Flow chart of the selection process.**

related to the best practices in vestibular rehabilitation [32] and motor imagery [33] for a comprehensive assessment of the included studies.

## Results

The flow chart of the selection process is illustrated in Fig 1. A total of 2543 items, including duplicates, were identified. After removing the duplicates, 2404 citations remained for the selection process. 1 more article was included from hand-searching. After the titles and abstracts stage, the initial search yielded 11 articles eligible for full-text screening. Only 10

were available in full-text; the last one was a protocol in progress. Of these articles, two met all the inclusion criteria and were selected for data extraction.

## Characteristics of the studies

All extracted data are shown in Table 1: extraction grid. One study was conducted in the United Kingdom [23], and the other in Germany [24]. Both were controlled trials (in one, participants were their own control), but no information was provided about randomization. The sample sizes were 10 and 26.

## Characteristics of the intervention

Measurements were taken immediately [23], or at days 3–6 after symptom onset [24]. The comparator was either a type of motor imagery incongruent/congruent with physical practice [23] or physical practice and motor imagery practice [24]. Videonystamography was used to evaluate the vestibulo-ocular reflex (VOR) and slow-phase velocity (SPV). This tool has been validated for use in these populations. The same type of imagery was used in both studies: whole-body, first-person kinesthetic motor imagery.

## Characteristics of the participants

Nigmatullina et al. included healthy subjects who underwent stimulation to simulate a vestibular disorder (mean age = 20.9 years, age range = 19–23 years, nine (experiment 1) + seven (experiment 2) females), whereas Jahn et al. included people with neuritis (mean age 51 years, range 30–74 years, two females).

## Critical appraisal of the studies

All extracted data are shown in Table 2: critical appraisal tool. A large proportion of data is missing. The data collected show low to no external validity, unrepresentative or poorly adapted assessment, lack of statistical validity, low internal validity, lack of use of motor imagery intervention recommendations, and lack of use of recommendations for practice in vestibular disorders. "See the supplementary material for explanations of the critical grid."

## Outcomes

Both studies measured the presence of nystagmus with videonystagmography. Data relating to the vestibular system were acquired via the VOR. Measurement of the VOR involves studying eye movements in response to vestibular stimulation. Vestibular nystagmus has a slow component triggered by vestibular signals and a fast, corrective component that causes movement in the opposite direction. The direction of the nystagmus is determined by that of its rapid component, which is easier to identify.

Both studies evaluated the SPV, a component of vestibular nystagmus. To measure the SPV, the distance traveled by the eye during the slow phase of the nystagmus is divided by the duration of the nystagmus (SPV = $\Delta\Theta/\Delta t$). The mean SPV was calculated for each nystagmus beat for each task, and results were displayed graphically. Both studies found a statistically significant reduction in SPV after the intervention. Nigmatullina et al. measured the SPV just after vestibular stimulation of rotatory origin, and Jahn et al. measured it 3–6 days after symptom onset. Neither study performed repeated measures after the first post intervention-evaluation.

## Imagery type

Nigmatullina et al. used implicit whole-body motor imagery (i.e., a whole-body rotation task), whereas Jahn et al. used explicit whole-body motor imagery (i.e., an imagined running task) The motor imagery modalities and perspectives were kinesthetic and first-person (i.e., sensorimotor sensation centered) in both studies. Neither study reported assessing the imagery skills of the participants.

**Table 2. Critical appraisal tool.**

| Critical appraisal criteria | Nigmatullina et al. | Jahn et al. |
|---|---|---|
| **External validity** | | |
| Representative sample | not representative low- | not representative low- |
| Material cost | not representative – high | appropriate – low |
| Assessment duration | | |
| Intervention duration | not representative – not known | adapted – low |
| feasibility element (tools, recruitment) | Yes | Yes |
| Mental imagery instruction | Yes | Yes |
| **Statistical validity** | | |
| Statistical power | – | – |
| | No sample size calculation | No sample size calculation |
| Fishing | ~ | ~ |
| | No published protocol | No published protocol |
| | No clear aims | No clear aims |
| **Internal validity** | | |
| Randomization | + | – |
| | Randomization of the intervention parameters | |
| Blinding | -no | -no |
| Control | -yes | -no |
| | But no information on the comparability of the two samples | |
| Evidence of measurement properties | – | – |
| | No information on validity | No information on validity |
| | No information on reliability | No information on reliability |
| **Recommendations for motor imagery practice** | | |
| Motor imagery capacity evaluation | – | – |
| | not known | not known |
| Perspective information | – | – |
| (internal or external) | not known | not known |
| Perspective evolution | – | – |
| | not known | not known |
| Modality information | – | – |
| (visual or kinesthesic) | not known | not known |
| Modality evolution | – | – |
| | not known | not known |
| Detailed MI instructions | – | – |
| | not known | not known |
| Familiarisation | – | – |
| | not known | not known |
| Closed eyes | – | – |
| | not known | not known |
| **Recommendations for vestibular disorders practice** | | |
| Rotary chair | not known | not known |
| Optokinetic stimulation | not known | not known |
| Vestibular stimulation exercises: | not known | not known |
| Vestibular habituation exercises: | not known | not known |
| Balance retraining exercises: | not known | not known |
| Eyes movement exercises | not known | not known |
| Head and eye coordination | not known | not known |

## Discussion

The aims of this study were 1. to identify motor imagery interventions used in the literature, 2. to critically appraise them, and 3. to report their impact on clinical outcomes. This systematic search and review identified two studies evaluating motor imagery: one in people with neuritis and the other in healthy people who underwent vestibular stimulation. Both studies had many limitations and risks of bias. Our findings suggest a possible effect of motor imagery on the vestibulo-ocular reflex (VOR), especially the SPV. However, several concerns were raised regarding the methodological choices and feasibility of the motor imagery interventions used in the included studies.

Although recommendations have been made regarding the use of motor imagery as part of vestibular rehabilitation [10], our review found little evidence to support this [23,24]. In addition to the fact only two studies have been published on this specific subject, the quality of the studies is limited. Only one study was controlled (in one, participants were their own controls), and it involved healthy individuals who underwent vestibular stimulation to simulate pathology. This method does not represent all vestibular pathologies and does not take into account all the dimensions of the pathological experience. The other study involved individuals with vestibular pathology, but the design did not allow the intervention to be controlled. Therefore, contrary to Lacour and Bernard-Demanze's proposal [10], it does not seem appropriate to recommend using motor imagery in clinical practice as a first-line treatment based on the scientific literature.

However, motor imagery could positively impact people living with vestibular disorders. These two studies suggest a significant effect of motor imagery on the VOR, especially the SPV, which is altered in all vestibular disorders [23,24]. This is a decisive point in our review, as nystagmographic measurement and quantification of this reflex are considered the gold standard in the international indexed literature [34,35]. Videonystagmography evaluates the vestibular system in a reproducible and non-invasive manner. Nevertheless, several limitations were identified. Firstly, the external validity of the studies is limited by the use of videonystagmography alone (nystagmography, although useful, does not assess all the components of the vestibular system individually, and does not take into account compensatory strategies implemented by the individual) [36,37], and the absence of pathology and the high cost of the intervention and evaluation for one [24] of the studies. Internal validity is also limited by the lack of blinding, the absence of a control arm in one study and a non-explicit random allocation in the other, as well as the lack of evidence supporting the outcome measures used [31]. Finally, the motor imagery rehabilitation programs used cannot be reproduced or optimized since no description of the motor imagery modalities or perspectives used was provided, and there was no motor imagery quality evaluation or progression of the program [33].

From a clinical point of view, the results suggest a potential benefit of using motor imagery as part of vestibular rehabilitation. However, these results need to be weighed against methodological considerations. Firstly, the number of studies selected, and the number of subjects involved in these studies was small (total of 36). Secondly, the representativeness of the samples is questionable. Although Jahn et al. evaluated individuals with vestibular pathology, this was limited to neuritis, which does not represent the diversity of pathophysiological processes involved in vestibular disorders. Nigmatulina et al. evaluated healthy subjects who underwent stimulation to reproduce a symptom, which also does not represent all vestibular diseases. Regarding the question posed in this review about recommendations for using motor imagery in vestibular rehabilitation [10], the limited data in the literature does not support such a recommendation despite some consistent findings in fundamental research [14–16].

However, from a research point of view, these results, together with the large body of theoretical models and literature showing the impact of vestibular diseases on the quality of motor imagery, suggest that studies should continue to investigate the clinical application of this method. Specifically, future research should include: 1. assessment of subjects' motor imaging quality (vestibular damage may affect imaging quality); 2. a battery of vestibular tests (Video Head Impulse Test assessment, functional impact and quality of life as measured by the Dizziness Handicap Inventory); 3. a clear, reproducible, and realistic rehabilitation program for use in routine practice, in accordance with recommendations on motor imaging [33] and vestibular rehabilitation [6]. For example, it should include motor imagery exercises related to the practice of

vestibular rehabilitation therapists, with imagery reproduction of these exercises, of a specific duration, intensity, and regularity. The number of subjects required should be determined based on a primary endpoint predefined before the study. The study must be multicenter in order to avoid a "center effect." Given the heterogeneity of vestibular pathologies, future studies should focus on a specific vestibular pathology (e.g., neuritis, Meniere's syndrome, etc.) to test these hypotheses. Finally, studies on healthy subjects receiving vestibular stimulation could be conducted.

The low quality of the studies included does not allow firm conclusions to be drawn regarding future research. Methodologically robust studies including representative samples of individuals with vestibular pathology are required. Comprehensive assessments should be conducted, as stated above. Studies should report a complete description of the motor imagery intervention, including an imagery quality assessment [38], the perspective and modality used to produce the image [11,33,39], prior familiarization, and the practice of imagery during or outside the sessions to enable a better understanding and reproducibility of the proposed rehabilitation program [11,33,39]. Finally, a long-term evaluation of the retention of the effect is necessary to determine the clinical benefit.

## Limitations

Differences in terminology across studies may have led to a loss of sensitivity in the research strategy. This review raises the question of the absence of clinical studies on this theme, with a possible undetected negative publication bias [40].

## Conclusion

This systematic search and review identified two studies evaluating the efficacy of motor imagery interventions for the rehabilitation of vestibular disorders. While studies suggest that motor imagery rehabilitation reduces the vestibulo-ocular reflex, there are some methodological concerns included the small sample size, absence of imagery quality assessment, and poor generalizability. However, although motor imagery appears to be affected by vestibular pathology, the paucity and lack of quality of the available evidence does not allow us to recommend its use in routine practice at this time.

## Supporting information

**S1 Appendix. Search strategy refinement.**
(DOCX)

**S2 Appendix. Additional material explaining the choice of analysis criteria in the critical grid.**
(DOCX)

## Acknowledgments

We thank Johanna Robertson, PT, PhD for English editing.

## Author contributions

**Conceptualization:** Dimitri Fabre-Adinolfi, Florian Naye, Thomas Rulleau.

**Data curation:** Dimitri Fabre-Adinolfi, Thomas Rulleau.

**Investigation:** Dimitri Fabre-Adinolfi, Thomas Rulleau.

**Methodology:** Dimitri Fabre-Adinolfi, Florian Naye, Thomas Rulleau.

**Supervision:** Thomas Rulleau.

**Validation:** Dimitri Fabre-Adinolfi, Florian Naye, Thomas Rulleau.

**Writing – original draft:** Dimitri Fabre-Adinolfi, Thomas Rulleau.

**Writing – review & editing:** Dimitri Fabre-Adinolfi, Florian Naye, Thomas Rulleau.

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
