## [Decision Letter · Decision Letter 0]

27 Oct 2025

Dear Dr. Rulleau,

Thank you for submitting your manuscript to PLOS ONE. After careful consideration, we feel that it has merit but does not fully meet PLOS ONE’s publication criteria as it currently stands. Therefore, we invite you to submit a revised version of the manuscript that addresses the points raised during the review process.

**ACADEMIC EDITOR:Please see the reviewer's comments, respond and revise the manuscript as needed.**

We look forward to receiving your revised manuscript.

Kind regards,

Gauri Mankekar, MD,PhD,FACS

Academic Editor

PLOS ONE

Additional Editor Comments (if provided):

Reviewers' comments:

Reviewer's Responses to Questions

**Comments to the Author**

1. Is the manuscript technically sound, and do the data support the conclusions?

Reviewer #1: Partly

Reviewer #2: Yes

2. Has the statistical analysis been performed appropriately and rigorously?

Reviewer #1: No

Reviewer #2: Yes

3. Have the authors made all data underlying the findings in their manuscript fully available?

Reviewer #1: Yes

Reviewer #2: Yes

4. Is the manuscript presented in an intelligible fashion and written in standard English?

Reviewer #1: Yes

Reviewer #2: Yes

Reviewer #1: The manuscript follows a systematic search and review approach, which is appropriate for emerging or under-researched topics but here few comments to consider

Explicitly compare “systematic search and review” vs. systematic review, and explain why GRADE or equivalent was not applied

The appraisal tool is not benchmarked against established checklists (e.g., ROBINS-I, Cochrane Risk of Bias).

No quantitative synthesis or effect-size estimation; e conclusions as underpowered.

Reviewer #2: Recommendation

The manuscript is relevant, timely, and methodologically transparent. However, before publication, the authors should:

Provide a more explicit assessment of certainty in evidence (even if narrative).

Clarify and justify the language restrictions.

Reframe conclusions with stronger caution regarding clinical applicability.

Improve clarity in reporting of intervention details and future research needs.

If these revisions are addressed, the manuscript would represent a valuable contribution to the literature on vestibular rehabilitation and motor imagery.

**Do you want your identity to be public for this peer review?** For information about this choice, including consent withdrawal, please see our Privacy Policy

Reviewer #1: **Yes:**  Ahmed Ibrahim Al Kharusi

Reviewer #2: No

---

## [Author Response · Author response to Decision Letter 1]

6 Nov 2025

Dear colleague, we hope we have answered all the experts' questions. Thank you for your interest in our work.

editor comments : Please see the reviewer's comments, respond and revise the manuscript as needed.

we hope we have answered all the experts' questions.

Reviewer 1

1 The manuscript follows a systematic search and review approach, which is appropriate for emerging or under-researched topics but here few comments to consider

We appreciate your interest and hope to clarify your request.

2 Explicitly compare “systematic search and review” vs. systematic review, and explain why GRADE or equivalent was not applied

We modified the manuscript accordingly.

Lines 100 to 105

“Unlike traditional systematic reviews, which aim to evaluate intervention effectiveness and therefore apply tools such as GRADE to rate the certainty of effect estimates, a systematic search and review focuses on describing interventions, examining their clinical applicability, and identifying conceptual or methodological gaps (https://onlinelibrary.wiley.com/doi/full/10.1111/j.1471-1842.2009.00848.x ; https://pubmed.ncbi.nlm.nih.gov/40914296/). In this context, GRADE or equivalent frameworks, which are designed to assess confidence in pooled or comparative effect estimates, are not appropriate (https://www.sciencedirect.com/science/article/pii/S2213398423002713)..”

3 The appraisal tool is not benchmarked against established checklists (e.g., ROBINS-I, Cochrane Risk of Bias).

We modified the manuscript accordingly.

A detailed explanation of this methodological choice is available below:

Because our review is a systematic search and review rather than a systematic review of intervention effectiveness, our purpose was not to estimate causal effects or formally compare trial outcomes. Instead, our objectives were to (1) describe the vestibular interventions used in clinical trials; (2) assess their clinical applicability and methodological limitations using a critical grid grounded in best practice in motor imagery and vestibular rehabilitation; and (3) summarize trial outcomes only to illustrate that current recommendations may overinterpret the evidence base.

For this type of big-picture review, established risk-of-bias tools such as ROBINS-I or RoB-2, which focus on internal validity for causal inference, are not appropriate or required. They do not evaluate intervention content, clinical applicability, or adherence to best practice principles. Our tailored appraisal grid is therefore aligned with the methodological intent of systematic search and review approaches and with our research question.

Lines 145 to 151

“One objective of a systematic search and review is to provide practice-oriented insights rather than to appraise internal validity for causal inference (https://onlinelibrary.wiley.com/doi/full/10.1111/j.1471-1842.2009.00848.x). Accordingly, we developed a tailored critical appraisal tool to evaluate the external validity, feasibility, and alignment of intervention components with current best practice in motor imagery and vestibular rehabilitation. This grid was designed to capture aspects of intervention content and clinical applicability that are not addressed by traditional risk-of-bias tools. Key elements of contemporary motor imagery and vestibular practice were therefore integrated to support a clinically meaningful interpretation of the included trials.”

4 No quantitative synthesis or effect-size estimation; conclusions as underpowered.

We did not conduct a quantitative synthesis because it was not aligned with the objectives of this systematic search and review. Although quantitative synthesis without meta-analysis is feasible, for example using the SWiM framework, such approaches remain appropriate only when the aim is to summarise effect estimates across studies. In our case, the primary objective was to describe intervention content, assess clinical applicability, and examine alignment with best practice in motor imagery and vestibular rehabilitation, rather than to quantify treatment effects. With only two included studies and substantial differences in intervention content, participants, and outcome constructs, applying a structured quantitative synthesis would have added little interpretive value and risked overstating the strength of the evidence. The absence of quantitative synthesis therefore does not render our conclusions “underpowered”; instead, it reflects the limited and heterogeneous evidence base and the practice-oriented focus of this review.

Reviewer 2

1 The manuscript is relevant, timely, and methodologically transparent. However, before publication, the authors should:

We would like to thank reviewer 2 for their comments on the value of our work.

2 Provide a more explicit assessment of certainty in evidence (even if narrative).

Table 2 illustrated the assesment. We had pointed out that a lot of data was missing. We have added a more explicit assessment, as requested.

Lines 192 -195 :

« The data collected show low to no external validity, unrepresentative or poorly adapted assessment, lack of statistical validity, low internal validity, lack of use of motor imagery intervention recommendations, and lack of use of recommendations for practice in vestibular disorders.”

3 Clarify and justify the language restrictions.

We restricted our search to studies published in English and French. These are the working languages of the review team, allowing accurate screening, data extraction, and interpretation without requiring external translation resources. Although the Cochrane Handbook recommends including studies in all languages to minimise selection bias, applying this principle requires access to reliable translation capacity. Using automated translation technologies may appear to overcome this barrier, but it carries risks of mistranslation, particularly for clinically nuanced terminology, intervention components, and methodological details, which could compromise the accuracy of data extraction and appraisal. In vestibular rehabilitation and motor imagery, where the vast majority of trials are published in English, this restriction is unlikely to have materially affected study retrieval or the conclusions.

4 Reframe conclusions with stronger caution regarding clinical applicability.

We appreciate your comment and have amended the conclusion accordingly.

Lines 283-287 :

“This systematic search and review identified two studies evaluating the efficacy of motor imagery interventions for the rehabilitation of vestibular disorders. While studies suggest that motor imagery rehabilitation reduces the vestibulo-ocular reflex, there are some methodological concerns included the small sample size, absence of imagery quality assessment, and poor generalizability. However, although motor imagery appears to be affected by vestibular pathology, the paucity and lack of quality of the available evidence does not allow us to recommend its use in routine practice at this time.”

5 Improve clarity in reporting of intervention details and future research needs.

We appreciate your comment and have amended the conclusion accordingly.

Lines 263 to 271 :

« Specifically, future research should include: 1. assessment of subjects' motor imaging quality (vestibular damage may affect imaging quality); 2. a battery of vestibular tests (Video Head Impulse Test assessment, quality of life as measured by the Dizziness Handicap Inventory); 3. a clear, reproducible, and realistic rehabilitation program for use in routine practice, in accordance with recommendations on motor imaging [29] and vestibular rehabilitation [6]. For example, it should include motor imagery exercises related to the practice of vestibular rehabilitation therapists, with imagery reproduction of these exercises, of a specific duration, intensity, and regularity. The number of subjects required should be determined based on a primary endpoint predefined before the study. The study must be multicenter in order to avoid a “center effect. Given the heterogeneity of vestibular pathologies, future studies should focus on a specific vestibular pathology (e.g., neuritis, Meniere's syndrome, etc.) to test these hypotheses. Finally, studies on healthy subjects receiving vestibular stimulation could be conducted.”.

6 If these revisions are addressed, the manuscript would represent a valuable contribution to the literature on vestibular rehabilitation and motor imagery

We would like to thank reviewer 2 for their comments on the contribution of our work on vestibular rehab and motor imagery

---

## [Decision Letter · Decision Letter 1]

9 Dec 2025

Thank you for submitting your manuscript to PLOS ONE. After careful consideration, we feel that it has merit but does not fully meet PLOS ONE’s publication criteria as it currently stands. Therefore, we invite you to submit a revised version of the manuscript that addresses the points raised during the review process.

We look forward to receiving your revised manuscript.

Kind regards,

Gauri Mankekar, MD,PhD,FACS

Academic Editor

PLOS One

Journal Requirements:

**Additional Editor Comments:**

The manuscript will be strengthened if the authors consider the following points.

1. Authors provide a link to the search criteria used (supplemental material to their PROSPERO submission), which also includes results, in terms of number of articles identified from a search conducted in August 2023. It appears for this manuscript, authors redid the search in June 2025. However, the number of articles identified using the same search criteria, in some cases, are quite a bit less than that identified in 2023 which does not make sense to me (for example, in 2023, PubMed identified 201 articles, but according to Figure 1, the current search only identified 48 from PubMed; similarly in 2023, CINAHL identified 33 articles, while the current search identified 9 and Cochrane library identified 70 in 2023 and only 5 in 2025). Authors should clarify reasons why the current search resulted in fewer articles from the same sources using the same criteria. If criteria were modified, that needs to be clarified.

2. Given that authors created their own set of criteria for the clinical evaluation of the interventions, authors should provide in supplemental material what each criterion represents and what information they were looking for (give examples).

Minor points:

1. line 75: there is a missing "." at the end of the sentence (between "functions" and "Yardley").

2. line 82: change "affect" to "affects"

3. line 113: there is an error with the reference (Error! Reference source not found)

4. line 164: "1 more articles" should be "1 more article"

5. line 183: authors mention 9 females in the Nigmatullina study, but this does not match what is presented in Table 1 - looking up the reference, it appears as though there were 2 experiments with different number of females in each experiment - it may be that authors only utilized information from 1 experiment, but that should be clarified.

Reviewers' comments:

Reviewer's Responses to Questions

**Comments to the Author**

Reviewer #3: (No Response)

2. Is the manuscript technically sound, and do the data support the conclusions?

Reviewer #3: Partly

3. Has the statistical analysis been performed appropriately and rigorously?

Reviewer #3: N/A

4. Have the authors made all data underlying the findings in their manuscript fully available?

Reviewer #3: Yes

5. Is the manuscript presented in an intelligible fashion and written in standard English?

Reviewer #3: Yes

Reviewer #3: The authors present a systematic search and review of motor imagery interventions used as part of vestibular rehabilitation, somewhat in response to an article that encouraged the use of motor imagery in rehabilitation. Authors felt this recommendation was potentially premature and wanted to review what information existed about the use of motor imagery. With only 2 studies identified that met their inclusion criteria, authors evaluated the information contained in the publications regarding the intervention and quality of evidence using a set of criteria they identified as being relevant and important. The overall conclusion was that evidence was not yet present to support active use of motor imagery in rehabilitation, but encouraged more research to be done to evaluate its impact on clinical measures. The manuscript will be strengthened if the authors consider the following points.

1. Authors provide a link to the search criteria used (supplemental material to their PROSPERO submission), which also includes results, in terms of number of articles identified from a search conducted in August 2023. It appears for this manuscript, authors redid the search in June 2025. However, the number of articles identified using the same search criteria, in some cases, are quite a bit less than that identified in 2023 which does not make sense to me (for example, in 2023, PubMed identified 201 articles, but according to Figure 1, the current search only identified 48 from PubMed; similarly in 2023, CINAHL identified 33 articles, while the current search identified 9 and Cochrane library identified 70 in 2023 and only 5 in 2025). Authors should clarify reasons why the current search resulted in fewer articles from the same sources using the same criteria. If criteria were modified, that needs to be clarified.

2. Given that authors created their own set of criteria for the clinical evaluation of the interventions, authors should provide in supplemental material what each criterion represents and what information they were looking for (give examples).

Minor points:

1. line 75: there is a missing "." at the end of the sentence (between "functions" and "Yardley").

2. line 82: change "affect" to "affects"

3. line 113: there is an error with the reference (Error! Reference source not found)

4. line 164: "1 more articles" should be "1 more article"

5. line 183: authors mention 9 females in the Nigmatullina study, but this does not match what is presented in Table 1 - looking up the reference, it appears as though there were 2 experiments with different number of females in each experiment - it may be that authors only utilized information from 1 experiment, but that should be clarified.

**Do you want your identity to be public for this peer review?** For information about this choice, including consent withdrawal, please see our Privacy Policy

Reviewer #3: No

---

## [Author Response · Author response to Decision Letter 2]

19 Dec 2025

The manuscript will be strengthened if the authors consider the following points.

Response : We thank the academic editor for this comment and for his interest in our work.

general comment :

1. Authors provide a link to the search criteria used (supplemental material to their PROSPERO submission), which also includes results, in terms of number of articles identified from a search conducted in August 2023. It appears for this manuscript, authors redid the search in June 2025. However, the number of articles identified using the same search criteria, in some cases, are quite a bit less than that identified in 2023 which does not make sense to me (for example, in 2023, PubMed identified 201 articles, but according to Figure 1, the current search only identified 48 from PubMed; similarly in 2023, CINAHL identified 33 articles, while the current search identified 9 and Cochrane library identified 70 in 2023 and only 5 in 2025). Authors should clarify reasons why the current search resulted in fewer articles from the same sources using the same criteria. If criteria were modified, that needs to be clarified.

Response : Thank you for this comment. Here is the justification provided in the newly developed supplementary material.

The review protocol was registered a priori in 2023. The initial search strategy for the motor imagery component was adapted from the only previously published review on this topic and reflected the terminology and database indexing available at the time of that review (Schuster, C., Hilfiker, R., Amft, O. et al. Best practice for motor imagery: a systematic literature review on motor imagery training elements in five different disciplines. BMC Med 9, 75 (2011). https://doi.org/10.1186/1741-7015-9-75).

The extended duration of the review required updating the searches to ensure currency. Given the low number of studies included and the evolution of terminology and database indexing since the 2011 review, we refined the search syntax to better align with current indexing practices and terminology, while preserving the original conceptual framework of the search and all protocol-defined eligibility criteria.

These refinements did not constitute a change in the research question, scope, or review methods. Rather, they represent an operational optimization of how predefined concepts were translated into database-specific search syntax. We deliberately chose not to amend the registered protocol, as the review methods remained unchanged, and the review had progressed beyond study selection and data extraction. Retrospective protocol modification to reflect search syntax refinements alone could introduce ambiguity regarding selective methodological adaptation.

The refined search strategy retrieved fewer records than the initial strategy, reflecting improved specificity rather than reduced coverage. All records retrieved across search iterations were deduplicated and screened using identical eligibility criteria and procedures. The refined strategy did not identify additional eligible studies beyond those already included, suggesting that the initial strategy had already achieved adequate coverage of the available evidence base and that the refinements served to confirm, rather than alter, the review findings.

To ensure transparency and reproducibility, the full search strategies for each database and search iteration are reported in the supplementary material.

Questions on this point led us to review our update as well. We apologize for the error that occurred during the last submission, where one of the databases was not queried correctly. We have made the correction and verified everything. There is therefore a discrepancy of 60 references with this correction.

Line 144: We added: “See Supplementary Material 1 for the full search strategy and justification of search strategy refinements following protocol registration

Line 45 : “2404”

Line 165 : 2543

Line 166 :“2404”

2. Given that authors created their own set of criteria for the clinical evaluation of the interventions, authors should provide in supplemental material what each criterion represents and what information they were looking for (give examples).

Response : We thank you for your comments. We have added supplementary material on these points, which will give consistency to our work.

Line 191: we added : “See the supplementary material for explanations of the critical grid.”

specific comment :

1. line 75: there is a missing "." at the end of the sentence (between "functions" and "Yardley").

Response : thanks

Line 75 : We added a dot between “functions” and “Yardley”

2. line 82: change "affect" to "affects" thanks

Response : Line 82: we changed "affect" to "affects"

3. line 113: there is an error with the reference (Error! Reference source not found)

Response : We have no “error” on our version. Il could be a compatibility problem.

Ref 23 : 23. Schiavenato M, Chu F. PICO: What it is and what it is not. Nurse Educ Pract. 2021;56: 103194. doi:10.1016/j.nepr.2021.103194

5. line 183: authors mention 9 females in the Nigmatullina study, but this does not match what is presented in Table 1 - looking up the reference, it appears as though there were 2 experiments with different number of females in each experiment - it may be that authors only utilized information from 1 experiment, but that should be clarified.

Response : Thanks

Line 183, we added “nine (experiment 1) + seven (experiment 2)”, in order to clarify.

The authors present a systematic search and review of motor imagery interventions used as part of vestibular rehabilitation, somewhat in response to an article that encouraged the use of motor imagery in rehabilitation. Authors felt this recommendation was potentially premature and wanted to review what information existed about the use of motor imagery. With only 2 studies identified that met their inclusion criteria, authors evaluated the information contained in the publications regarding the intervention and quality of evidence using a set of criteria they identified as being relevant and important. The overall conclusion was that evidence was not yet present to support active use of motor imagery in rehabilitation, but encouraged more research to be done to evaluate its impact on clinical measures. The manuscript will be strengthened if the authors consider the following points.

Response : We thank the reviewer for this comment and for his interest in our work.

general comment :

1. Authors provide a link to the search criteria used (supplemental material to their PROSPERO submission), which also includes results, in terms of number of articles identified from a search conducted in August 2023. It appears for this manuscript, authors redid the search in June 2025. However, the number of articles identified using the same search criteria, in some cases, are quite a bit less than that identified in 2023 which does not make sense to me (for example, in 2023, PubMed identified 201 articles, but according to Figure 1, the current search only identified 48 from PubMed; similarly in 2023, CINAHL identified 33 articles, while the current search identified 9 and Cochrane library identified 70 in 2023 and only 5 in 2025). Authors should clarify reasons why the current search resulted in fewer articles from the same sources using the same criteria. If criteria were modified, that needs to be clarified.

Response : Thank you for this comment. Here is the justification provided in the newly developed supplementary material.

The review protocol was registered a priori in 2023. The initial search strategy for the motor imagery component was adapted from the only previously published review on this topic and reflected the terminology and database indexing available at the time of that review (Schuster, C., Hilfiker, R., Amft, O. et al. Best practice for motor imagery: a systematic literature review on motor imagery training elements in five different disciplines. BMC Med 9, 75 (2011). https://doi.org/10.1186/1741-7015-9-75).

The extended duration of the review required updating the searches to ensure currency. Given the low number of studies included and the evolution of terminology and database indexing since the 2011 review, we refined the search syntax to better align with current indexing practices and terminology, while preserving the original conceptual framework of the search and all protocol-defined eligibility criteria.

These refinements did not constitute a change in the research question, scope, or review methods. Rather, they represent an operational optimization of how predefined concepts were translated into database-specific search syntax. We deliberately chose not to amend the registered protocol, as the review methods remained unchanged, and the review had progressed beyond study selection and data extraction. Retrospective protocol modification to reflect search syntax refinements alone could introduce ambiguity regarding selective methodological adaptation.

The refined search strategy retrieved fewer records than the initial strategy, reflecting improved specificity rather than reduced coverage. All records retrieved across search iterations were deduplicated and screened using identical eligibility criteria and procedures. The refined strategy did not identify additional eligible studies beyond those already included, suggesting that the initial strategy had already achieved adequate coverage of the available evidence base and that the refinements served to confirm, rather than alter, the review findings.

To ensure transparency and reproducibility, the full search strategies for each database and search iteration are reported in the supplementary material.

Questions on this point led us to review our update as well. We apologize for the error that occurred during the last submission, where one of the databases was not queried correctly. We have made the correction and verified everything. There is therefore a discrepancy of 60 references with this correction.

Line 144: We added: “See Supplementary Material 1 for the full search strategy and justification of search strategy refinements following protocol registration

Line 45 : “2404”

Line 165 : 2543

Line 166 :“2404”

2. Given that authors created their own set of criteria for the clinical evaluation of the interventions, authors should provide in supplemental material what each criterion represents and what information they were looking for (give examples).

Response : We thank you for your comments. We have added supplementary material on these points, which will give consistency to our work.

Line 191: we added : “See the supplementary material for explanations of the critical grid.”

specific comment :

1. line 75: there is a missing "." at the end of the sentence (between "functions" and "Yardley").

Response : thanks

Line 75 : We added a dot between “functions” and “Yardley”

2. line 82: change "affect" to "affects" thanks

Response : Line 82: we changed "affect" to "affects"

3. line 113: there is an error with the reference (Error! Reference source not found)

Response : We have no “error” on our version. Il could be a compatibility problem.

Ref 23 : 23. Schiavenato M, Chu F. PICO: What it is and what it is not. Nurse Educ Pract. 2021;56: 103194. doi:10.1016/j.nepr.2021.103194

5. line 183: authors mention 9 females in the Nigmatullina study, but this does not match what is presented in Table 1 - looking up the reference, it appears as though there were 2 experiments with different number of females in each experiment - it may be that authors only utilized information from 1 experiment, but that should be clarified.

Response : Thanks

Line 183, we added “nine (experiment 1) + seven (experiment 2)”, in order to clarify.

---

## [Editor Report · Decision Letter 2]

23 Dec 2025

Evidence for motor imagery in the management of vestibular disorders does not support recent guidelines: a systematic search and review

PONE-D-25-36279R2

Dear Dr. Rulleau,

We’re pleased to inform you that your manuscript has been judged scientifically suitable for publication and will be formally accepted for publication once it meets all outstanding technical requirements.

Kind regards,

Gauri Mankekar, MD,PhD,FACS

Academic Editor

PLOS One
---

## [Editor Report · Acceptance letter]

PONE-D-25-36279R2

PLOS One

Dear Dr. Rulleau,

I'm pleased to inform you that your manuscript has been deemed suitable for publication in PLOS One. Congratulations! Your manuscript is now being handed over to our production team.

Kind regards,

on behalf of

Dr. Gauri Mankekar

Academic Editor

PLOS One